behaviour

social foraging, resource competition,
state-space model

**Author for correspondence:**
Tatsuya Kameda
e-mail: tkameda@l.u-tokyo.ac.jp

# Mere presence of co-eater automatically shifts foraging tactics toward 'Fast and Easy' food in humans

Yukiko Ogura[1], Taku Masamoto[1]
and Tatsuya Kameda[1,2,3]

[1]Department of Social Psychology, The University of Tokyo, Japan
[2]Brain Science Institute, Tamagawa University, Tokyo, Japan
[3]Center for Experimental Research in Social Sciences, Hokkaido University, Hokkaido, Japan

YO, 0000-0002-2702-2221

Competition for food resources is widespread in nature. The foraging behaviour of social animals should thus be adapted to potential food competition. We conjectured that in the presence of co-foragers, animals would shift their tactics to forage more frequently for smaller food. Because smaller foods are more abundant in nature and allow faster consumption, such tactics should allow animals to consume food more securely against scrounging. We experimentally tested whether such a shift would be triggered automatically in human eating behaviour, even when there was no rivalry about food consumption. To prevent subjects from having rivalry, they were instructed to engage in a 'taste test' in a laboratory, alone or in pairs. Even though the other subject was merely present and there was no real competition for food, subjects in pairs immediately exhibited a systematic behavioural shift to reaching for smaller food amounts more frequently, which was clearly distinct from their reaching patterns both when eating alone and when simply weighing the same food without eating any. These patterns suggest that behavioural shifts in the presence of others may be built-in tactics in humans (and possibly in other gregarious animals as well) to adapt to potential food competition in social foraging.

## 1. Introduction

Many animal species forage for food with other individuals. Social foraging provides animals with various benefits, including higher encounter rate with food resources and collective monitoring against predators, due to information sharing [1]. On the other hand, such information sharing also creates competition among

co-foragers, because others can free-ride on a food-finder's search efforts and steal their food [2]. To deal with such competition, animals may adopt different foraging tactics in a group situation than when foraging alone.

Much research has been conducted on social foraging from a macroscopic perspective, such as on the effects of group size [3]. Although quite a few studies in behavioural ecology have examined how individuals in a group adjust their foraging tactics depending on social contexts, their main scope is on the effects of predation risk [4] or social rank [5], and relatively few studies have focused on the effects of resource competition *per se*. Classical psychological studies have shown an increase in the frequency of eating behaviour [6,7] and/or food intake [6,8–10] in the presence of conspecifics, a phenomenon called 'social facilitation' [11,12] of eating behaviour. However, such psychological research (especially with humans) has generally lacked a tactical perspective.

How do animals shift their foraging tactics in a group? From the viewpoint of behavioural ecology, one possibility is that, to cope with possible competition for food from other group members, individuals may increase their frequency of approaching food patches, even though it increases overall foraging time and effort. A meta-analysis of primate studies revealed that foraging time generally grows with group size [13]. Another compatible hypothesis is that animals may choose smaller foods at the cost of quantity per intake. As smaller foods are easier to find [14,15] and require shorter consumption time, such a shift in preference should enable the finder to keep more of the food it finds. Indeed, research with crows has shown that smaller foods are less likely to be scrounged than larger foods [16].

We thus conjecture that a social foraging context may promote adoption of the behavioural tactic of retrieving 'smaller foods more frequently' rather than 'larger foods less frequently', because the former reduces the risk of scrounging by other individuals. If such behavioural pattern is an adaptation to the socio-ecological structure of resource competition, animals may have a built-in system that automatically triggers the tactical shift, even in the absence of actual food competition.

A previous study with domestic chicks, which have been known to exhibit strong sensitivities to the presence of conspecifics [17,18], provided some support for this conjecture. Ogura & Matsushima [19] revealed that the chicks approached and pecked for food more frequently when a co-forager was present, compared to when they were isolated. The frequency of approaching and pecking behaviour in paired chicks increased even when there was no real competition for food (i.e. they were separated from each other by a transparent wall). Because resource competition is a universal problem requiring an adaptive solution in social foraging, we hypothesize that such tactical shift may emerge in humans as well, which are phylogenetically distant but have been known to exhibit high social sensitivities to the presence of conspecifics [20].

In this study, we provided human subjects with a plate of potato chips and compared their foraging behaviour in a solo situation (*Solo* condition hereafter; figure 1*a* bottom) and a pair situation (*Visible Pair* condition; figure 1*a* top). To examine whether the behavioural shift would be triggered even in the absence of competition, each subject was provided with a separate plate of potato chips for 'tasting and rating' (i.e. there was no competition for consumption of the potato chips). Subjects in the *Visible Pair* condition could see each other's eating behaviour but were only allowed to eat from their own plates. We then recorded their frequencies of reaching for potato chips, the weight of potato chips taken at each reach and the total amount consumed. We expected that, compared to the *Solo* condition, subjects in the *Visible Pair* condition would exhibit higher reaching frequency and smaller amounts (in terms of weight) per reach.

We were also interested in which aspect of the hypothesized behavioural changes (increase in reach frequency or decrease in food amount per reach) might be a more primary component of the shift in tactics for social foraging. Given the common ecological trade-off between food size and search or consumption time required for the food (e.g. [21]), the increase in reach frequency and the preference for smaller food should be coupled together as a behavioural package. However, when we consider how such foraging tactics are implemented behaviourally or neurally, it is important to clarify which process may be more primary in the behavioural hierarchy. To address this issue, we created another experimental condition in which the degree of subjective competition was assumed to be intermediate between the *Solo* condition and the *Visible Pair* condition: in the *Invisible Pair* condition, pairs of subjects were separated by an opaque partition and were invisible (but audible) to each other during the test. We assumed that the partition, which separated the two subjects and the two food trays, should physically reduce the possibility of competition in the *Invisible Pair* condition, compared to the *Visible Pair* condition.

## 2. Material and methods

Subjects were randomly assigned to one of three conditions: *Solo* (n = 19, 9 males and 10 females,), *Invisible Pair* (n = 20, 10 males and 10 females, 10 pairs) and *Visible Pair* (n = 20, 10 males and 10 females, 10 pairs).

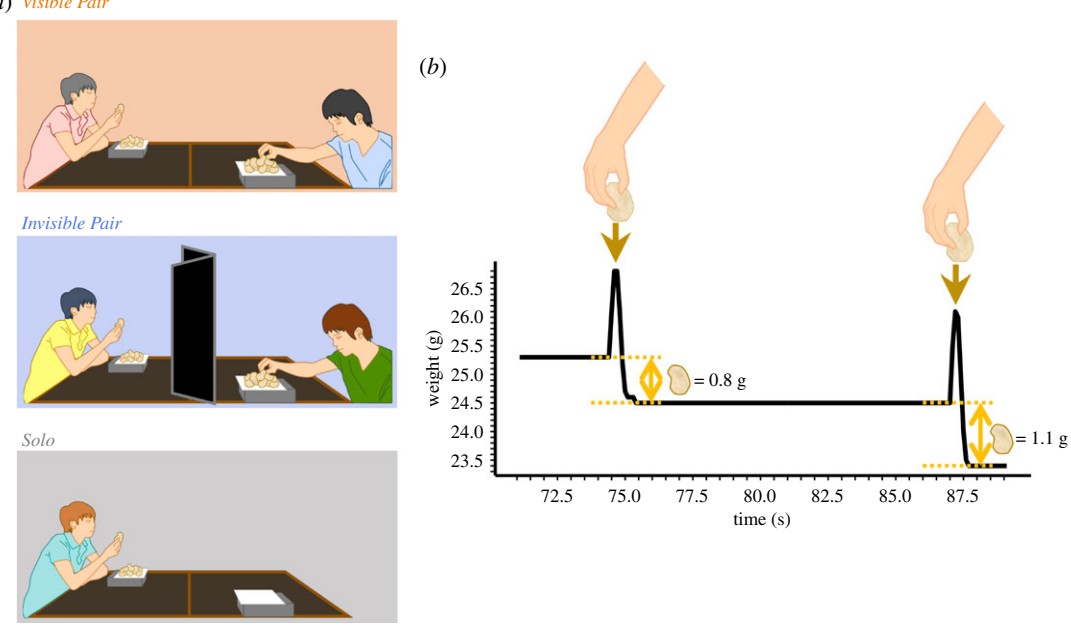

**Figure 1.** Schematic representations of experimental set-up and weight trajectory. (*a*) In the *Visible Pair* (top) and *Invisible Pair* (middle) conditions, paired subjects sat at opposite corners of a table (120 × 140 cm), each with their own plate of potato chips placed in front of them. In the *Invisible Pair* condition only, an opaque folding partition was placed between the two subjects. In the *Solo* condition (bottom), each subject sat alone at the table. (*b*) An example weight trajectory recorded by the electronic balance. The horizontal axis represents the elapsed time from the beginning of a session. For illustrative purposes, we show only a time window from 71 to 89 s with 0.5 s resolution (the actual time resolution of the balance used in the analyses was 0.1 s). Each reach was accompanied by a spike on the weight trajectory, so we measured the weight of potato chip intake for the reach as the difference between the weights before and after the spike.

Given that previous studies suggest social effect on eating behaviour could be different by sex [22,23], we conducted experiments separately for men and women. In both *Pair* conditions, two same-sex subjects who met for the first time at the experiment were seated at opposite corners of a table (figure 1*a*). In the *Visible Pair* condition, there was nothing that blocked subjects' view of each other. In the *Invisible Pair* condition, an opaque partition on the table separated the two subjects. Each subject was provided with a separate plate of potato chips. Subjects were instructed to 'taste and rate' the potato chips from their own plates (so that feelings of food competition were minimized). To examine whether saturation may affect eating in the presence of another co-eater, we repeated the test sessions three times in the experiment, each consisting of a 3 min (unspecified to subjects) tasting phase and a rating phase. At the beginning of the tasting phase, a 30 g portion of potato chips was placed on a separate plate in front of each subject. Subjects were also instructed that they could eat any amount at their own pace during the tasting phase. An electronic balance, which was invisible to subjects using a cardboard cover, was placed under the plate to record the timing and weight of potato chips taken in each reach (figure 1*b*). In the rating phase, subjects evaluated the potato chips on nine Likert scales (e.g. general deliciousness, saltiness and so on). After finishing the three sessions, subjects answered a post-session questionnaire that asked their subjective impressions of the taste test and general attitudes about eating behaviour. To ensure that subjects had not been satiated, we instructed them not to eat two hours before the experiment. For details, see Experimental procedures in the electronic supplementary material, materials and methods.

Separate from the main tasting experiment, we also asked two other naive female subjects to participate in a weighing experiment. There, as in the main experiment, a 30 g portion of potato chips was placed on a plate on the electronic balance. Each subject was asked to pick up chips one by one from the plate and to record each piece's weight (displayed as a weight decrement on the electric balance) sequentially on a sheet of paper. Each subject separately repeated this weighing task for 15 different plates. These data provided a behavioural baseline that allowed us to compare subjects' reaching pattern when eating potato chips with a situation that was physically similar but did not include eating.

We analysed reach frequency and total food intake by Bayesian multilevel regression. We constructed a linear model of reach frequency and total food intake with condition (*Visible Pair* or *Invisible Pair* or *Solo*), phase (1 to 3), sex (male versus female) and all the combinations of interactions as fixed effects.

As random effects, we specified subject-wise random intercepts, which were nested in pair-wise random intercepts, to account for repeated measures. Posterior distributions of the parameters have been obtained by Markov chain Monte Carlo (MCMC) sampling. The designed models were compared according to widely applicable information criteria (WAIC). Because WAIC varies depending on the random number seed of the MCMC, we ran 100 sampling runs per model with different seeds and counted how many times each model yielded minimum WAIC in all models. The model that yielded minimum WAIC highest number of times was regarded as the best model.

We analysed food intake per reach by state-space modelling. We assumed that the observed weight of potato chips in the $n$th reach was sampled from a latent state, which was in turn assumed to be sampled from the state in time unit $n − 1$. We also modelled the correlation within an individual as a random intercept. Then the differences between *Solo* (as a baseline) versus *Visible Pair* and *Invisible Pair* were included in the model.

To check the adequateness of the statistical models, we conducted posterior predictive check by comparing the prediction from the model and the observed data.

# 3. Results

## 3.1. Reach frequency increased in the *pair* conditions but not the overall amount of food intake

In the *Visible* and *Invisible Pair* conditions, reach frequencies of subjects were higher compared to the *Solo* condition (figure 2*a*, left). The best statistical model for reach frequency included *condition* as an explanatory variable and the 95% credible intervals (CIs) for the coefficients for *condition* were above 0 (electronic supplementary material, table S1; *Visible Pair* condition CI 0.09–0.78, *Invisible Pair* condition CI 0.14–0.84, compared with the *Solo* condition). This result supports our hypothesis of a behavioural shift toward more frequent access to food sources in a social setting. On the other hand, overall food intake did not increase in either the *Visible or Invisible Pair* condition as compared to the *Solo* condition (figure 2*a*, right). The best statistical model for the overall food intake amount did not include *condition* as an explanatory variable (electronic supplementary material, table S2).

Both for reach frequency and overall food intake, the best models contained sex as an explanatory variable. Both reach frequency and overall food intake were higher in men than women (electronic supplementary material, tables S1 and S2). This is in line with the questionnaire result that women were more apprehensive about the quantity of meals than men (CI 15.47–35.14; electronic supplementary material, figure S2 and table S3). However, the interaction *sex × condition* effect was not observed in either reach frequency or total food intake. That is, increase in reach frequency was on the same magnitude for both men and women.

The best model for reach frequency also contained *session* as an explanatory variable (electronic supplementary material, table S1). In the third session, subjects reached more frequently (CI 0.00–0.15) than in the first session. Furthermore, no interaction effect (*session × condition*, *sex × session*, and *sex × condition × session*) was observed. Taken together, these results imply that a saturation effect was not observed in this experiment.

While the experimental conditions affected subjects' eating behaviours, they had no effect on subjects' ratings about general deliciousness or other flavour and texture items for the taste of potato chips (electronic supplementary material, figure S1). Furthermore, neither reach frequency nor food intake amount predicted the rating about general deliciousness, i.e. higher reach frequency or food intake amount did not contribute to higher subjective rating about general deliciousness (electronic supplementary material, table S4).

## 3.2. Weight of potato chips per reach decreased immediately in the *visible pair* condition

The above patterns are in line with the prediction that mere presence of a co-eater will reduce the weight of food intake per reach (i.e. promote a preference for smaller foods). To confirm this point directly, we examined the temporal sequence of the weight of potato chip intake in each reach by each subject (figure 2*b*). State-space modelling (for details, see state-space modelling for the weight of potato chips per reach in the electronic supplementary material, materials and methods) revealed that, compared to the *Solo* condition, subjects in the *Visible Pair* condition selected smaller pieces even from their first reaches. As seen in figure 2*b*, the means ±95% CI of the *Visible Pair* condition were lower than the means of the *Solo* condition from the 1st until the 16th reach, indicating that the behavioural shift

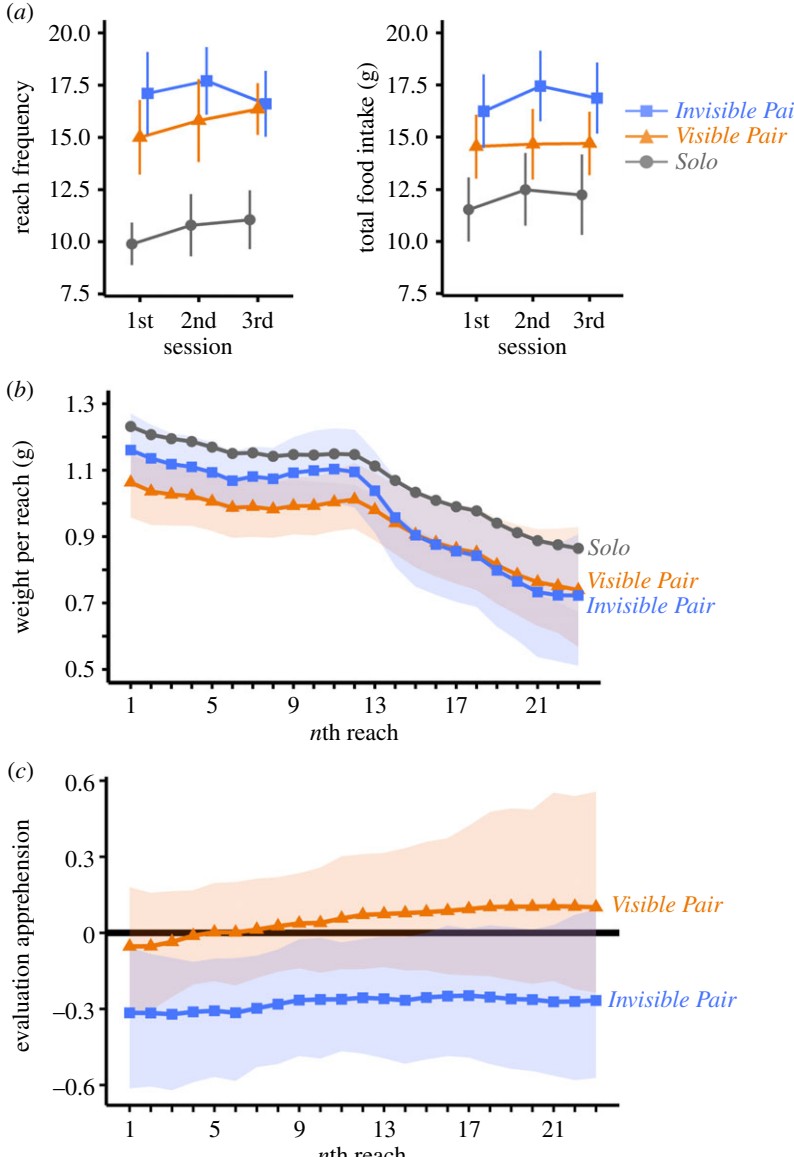

**Figure 2.** Increased reach frequency and preference for smaller food in the *Visible Pair* condition. (*a*) Group mean ± s.e.m. of reach frequency (left panel) and total food intake (right panel) versus session within the experiment. (*b*) Means of potato chip weight per reach in each condition as estimated by state-space modelling (see Material and methods for details). In the state-space model, we set the group mean of the *Solo* condition as a baseline and estimated the differences from that baseline for the *Visible Pair* and *Invisible Pair* conditions. Shaded orange and blue areas indicate 95% credible intervals (CIs) for the means of the *Visible Pair* and the *Invisible Pair* condition, respectively. (*c*) Estimated coefficients for evaluation apprehension during tasting. Shaded orange and blue areas indicate 95% credible intervals. Negative (i.e. lower than zero) value indicates that higher value of self-rated evaluation apprehension is associated with lower weight of potato chips per reach. When the shaded area contains zero, it indicates that, at that time point, the evaluation apprehension score does not explain the weight per reach.

toward 'foraging for smaller food' was immediately triggered when a co-eater was in sight. This immediate behavioural shift was not observed in the *Invisible Pair* condition. The means ±95% CI in the *Invisible Pair* condition differed from the means of the *Solo* condition only for the 15th to the 21st reach, suggesting that the behavioural shift toward smaller food amounts per reach was triggered to a lesser extent when the co-eater was not visible than when visible. Consistent with this interpretation, subjects in the *Invisible Pair* condition reported subjectively in the post-session questionnaire that they cared less about the presence of the co-eater during the tasting task, compared to those in the *Visible Pair* condition (95% CI: −2.91 to −0.94). These patterns indicate that the increase in reach frequency may be a more robust behavioural response to a social foraging situation (as observed in both *Pair* conditions) than the preference shift toward smaller food amounts per reach.

### 3.3. Social norm about eating behaviour does not explain the behavioural shift

From the perspective of human psychology, the decrease in the weight of potato chips per reach in the *Visible Pair* condition may be argued to be a result of 'evaluation apprehension' [24,25]. Subjects in the *Visible Pair* condition might be more apprehensive about being evaluated as excessive eaters. To test this possibility, we included a subjective rating score for apprehensiveness about the co-eater in the state-space model. This analysis revealed that, in the *Visible Pair* condition, subjective apprehensiveness did *not* explain the decrease in the weight of potato chips per reach; the 95% credible interval of the coefficient of score for apprehensiveness included 0 at all timepoints (figure 2*c*; 1st to 23rd reach). By contrast, in the *Invisible Pair* condition, higher apprehensiveness was related with smaller weight of potato chips per reach in earlier reaches (figure 2*c*; 1st to 14th reach). Thus, the effect of 'evaluation apprehension' was different between the *Visible Pair* condition and the *Invisible Pair* condition, i.e. subjective apprehensiveness partially explained the decrease in the weight of potato chips in the *Invisible Pair* condition, but not in the *Visible Pair* condition. Thus, social norm might have been at work partially, but could not explain the entire behavioural patterns (more frequent reach for smaller food) observed in this experiment.

### 3.4. Reach pattern in the visible pair condition was clearly distinguishable from random sampling of potato chips

The reaching pattern observed in figure 2*b* suggests that subjects in pairs selected smaller pieces of potato chips as compared to the solo situation. However, this can also be explained as a by-product, in which subjects were distracted by co-eaters and randomly sampled potato chips from the plate.

To assess this issue, we first weighed the contents of 100 bags of potato chips containing a total of 8075 pieces and investigated their weight distribution (figure 3*a*). Then, we artificially generated 60 sequences (comparable to the number of sequences in the main experiment) of random sampling from the weight distribution and compared the weight per reach between the random sequence and the actual sequences observed in the three conditions of the main (tasting) experiment. As seen in figure 3*b*, the choice pattern in the *Visible Pair* condition clearly deviated from the random sequence from the 1st reach to the 18th reach, indicating the subjects' choices were not random.

### 3.5. Reach pattern when eating potato chips was also clearly distinguishable from the reach pattern for (non-social) weighing

We also compared the time sequences of the reach patterns between the three conditions of the main (tasting) experiment and the weighing experiment. In the weighing experiment (figure 3*c* left), we observed how subjects chose potato chips over time when they reached for them only for the purpose of weighing (rather than for eating). Figure 3*c* right displays the results of state-space modelling of subjects' reach behaviours in the tasting experiment as deviated from the reach behaviours in the weighing experiment (0 means that there was no deviation from the baseline). As seen in figure 3*c*, the estimated deviation in the *Solo* condition was very close to 0 for all reaches. The 95% CI of estimated deviations in the *Invisible Pair* condition also included 0. However, the choice pattern in the *Visible Pair* condition deviated from the baseline from the 1st reach to the 14th reach, indicating the subjects' heightened preferences for smaller pieces. Given that the sample size was small ($n = 2$; see posterior predictive distribution in electronic supplementary material, figure S3) and that the experiment was conducted in a non-social situation, this result should be treated with some caution. Yet, the observed reach behaviours in the *Visible Pair* condition were distinguishable from the reach behaviours for (non-social) weighing.

## 4. Discussion

We created a laboratory foraging situation in which subjects were asked to eat potato chips for a 'taste test'. The mere presence of a co-eater in the *Visible Pair* condition increased the reach frequency for food and decreased the weight of food per reach, as compared to the *Solo* condition (figures 2*a* and *b*). This result supports our hypothesis that the behavioural shift toward foraging smaller food more frequently would be triggered automatically among human subjects, even when there was no actual competition about food consumption.

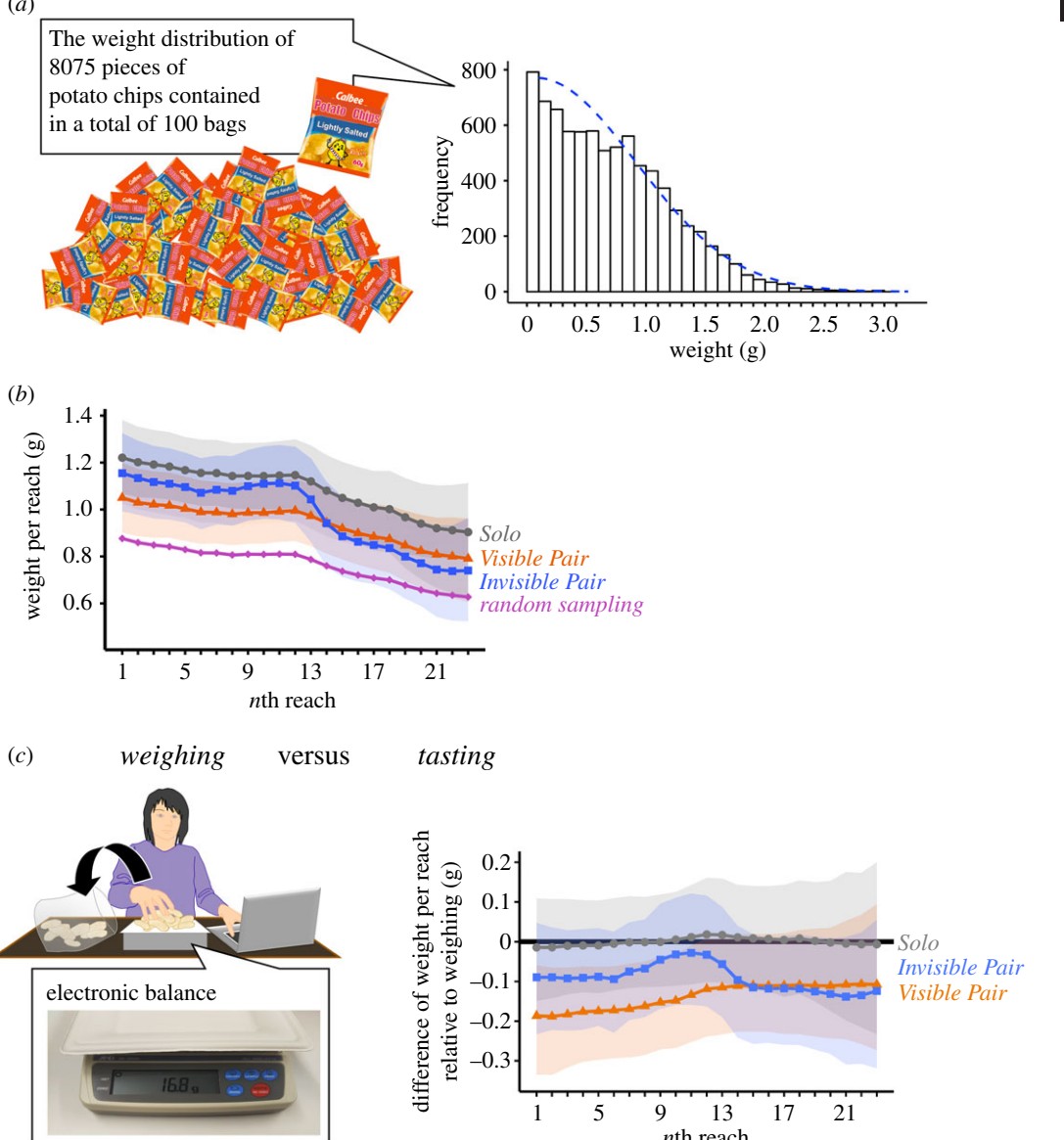

**Figure 3.** Experimental set-up and results of the weighing experiment (Study 1). (*a*) Histogram (bin size = 0.1) of the weight distribution of 8075 pieces of potato chips from one hundred 60 g packages. The dashed blue trace indicates the fitting curve of a skew-Gaussian model (mean = 0.74, s.d. = 0.50, skewness = 53.1). The weight distribution of potato chips used in Study 1 with human subjects displayed a similarity to biomass distribution in nature, where small foods are found relatively frequently (and easily), while larger foods are rarer [14,15]. (*b*) Estimated means of potato chip weight per reach in the main tasting experiment relative to the artificial weight sequence obtained by random sampling from the weight distribution of potato chips (displayed in figure 3*a*). Shaded areas indicate 95% credible intervals (CIs). (*c*) Estimated difference of potato chip weight per reach in the main tasting experiment (right), relative to the weighing experiment (left). Shaded areas indicate 95% credible intervals (CIs). Negative (less than 0) value indicates that the weight of potato chips taken was lower than that in the weighing experiment for the *n*th reach.

We argued that the behavioural tactics in social foraging consist of two components—increasing reach frequency and preferring smaller food amounts. Compared to the increase in reach frequency observed across the two *Pair* conditions, the behavioural shift for smaller food amounts emerged only in the *Visible Pair* condition. Although the latter shift may be seen as a by-product of random picking caused by distraction from the visible co-eater, the reach pattern was distinct from the simulated random sampling (figure 3*b*). It was also distinguishable from the counting pattern in the weighing experiment (figure 3*c*). We thus think that, along with increasing reach frequency, choosing smaller food amounts is a systematic (yet weaker) component of foraging tactics in human group settings.

The overall amount of individuals' food intake was not increased by the presence of a co-eater, which may appear inconsistent with results from previous human psychological studies of social facilitation in eating behaviour [10,26]. However, in these studies, food was freely given to the subjects and eating time was not controlled; in a subsequent study of real-world eating behaviour by humans, the increase in food consumption in the presence of others reflected an increase in meal duration [27]. While these psychological studies were silent about the cost–benefit trade-offs in foraging tactics, the present study examined human eating behaviour from a behavioural ecological perspective, arguing that humans may favour sure gain at the cost of time, effort and amount per intake to adjust to potential competition in social foraging.

Our results showed that the behavioural shift was triggered by the mere presence of a co-eater, even without actual competition. This suggests that the underlying mechanism for the shift may be a built-in system that activates automatically in response to relevant social cues. Considering that gregariousness is not human-specific but widespread in animals, neural implementation of an automatic competitive mode may also be rooted in ancient neural circuits. In domestic chicks, for example, a brain region considered to be homologous to the limbic area in mammals contributes to an automatic increase in the reaching frequency for feeders [28,29]. On the other hand, many brain mapping studies in humans have attempted to identify brain regions related to social competition using behavioural games [30–32]. However, the competitive contexts they have used are very different from the foraging situation in our study. Future research addressing the neural implementation of an automatic competitive mode in social foraging will be important not only for behavioural ecology but also to better understand the biological bases of problematic eating behaviour in humans.

In summary, humans shift their foraging tactics when a co-eater is present. Such a behavioural shift is likely to be a built-in response to possible food competition with conspecifics and may be common across many gregarious animals.

Ethics. The study was approved by, and carried out in accordance with the guidelines and regulations of, the ethical committee of the Department of Social Psychology of the University of Tokyo. All subjects responded to an advertisement for a 'potato chip tasting experiment', and provided written informed consent, approved by the ethical committee (IRB_SP2017_001), prior to the experiment.

Data accessibility. Data and analysis scripts are available within the Dryad Digital Repository: https://doi.org/10.5061/dryad.x95×69pcv [33].

Authors' contributions. Y.O., T.M. and T.K. designed the study. Y.O. and T.M. collected and analysed the data. Y.O. and T.K. wrote the paper. T.K. supervised the entire process of the study. All authors gave final approval for publication.

Competing interests. The authors declare no competing interests.

Funding. This study was supported by Japan Society for the Promotion of Science (JSPS) Grant-in-Aid for Scientific Research to T.K. (JP16H06324) and Y.O. (JP18K13267), and Japan Science and Technology Agency CREST Grant to T.K. (JPMJCR17A4-17941861). Support from CiSHub at the University of Tokyo is also appreciated.

Acknowledgements. We deeply appreciate Prof. Toshiya Matsushima's helpful comments. We thank Prof. Hiroshi Shimizu for advice on the statistical analyses.

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
