## [Reviewer comments · Royal Society Open Science]

Review History

RSOS-191598.R0 (Original submission)

Review form: Reviewer 1

Is the manuscript scientifically sound in its present form?

No

Are the interpretations and conclusions justified by the results?

No

Is the language acceptable?

Yes

Do you have any ethical concerns with this paper?

No

Have you any concerns about statistical analyses in this paper?

No

Recommendation?

Major revision is needed (please make suggestions in comments)

Comments to the Author(s)

This study investigates how the presence of another individual influences foraging behaviour in humans and domestic chicks. The main finding is that both species reach smaller amounts of food more frequently when a conspecific is present, and this is suggested to result from the perceived risk of competition. The manuscript is well written and easy to read, and the statistical analyses are appropriate. After getting feedback from their previous submission to Proceedings of the Royal Society B, authors have conducted an additional test that shows that the choice pattern of humans in the presence of another person deviates from random sampling. This makes the conclusions about the shift towards smaller amounts of food stronger in Study 1. However, I think that there are still some issues (listed below) that should be addressed before publication.

One of the main conclusions of the paper is that humans and chicks show identical behavioural patterns, however, I'm not convinced that Study 1 and 2 are totally comparable. In contrast to human subjects that could take different amounts of chips per reach, I assume that chicks never took more than one grain per peck? Fig. 3c shows the probability of taking a grain per peck, which suggests that the presence of a conspecific mainly reduced chicks' likelihood to take a grain, rather than shifting their strategy to reaching smaller amounts of food. I therefore think that conclusions about parallel results in two studies (and setup) should be toned down. To be able to better compare the two species, future studies could present chicks with grains of different sizes to see if the presence of a conspecific shifts their preferences towards smaller food items.

The main result from Study 2 is that the peck frequency of chicks increased in the presence of a conspecific, and authors interpret this to be a result of competition. However, an alternative explanation is that as social species, chicks in solitary condition may have been more stressed and vigilant and therefore spent less time foraging. The presence of a companion chick might have reduced this stress and witnessing others foraging could have facilitated the same behaviour in observers, therefore increasing peck frequency. I think that this alternative explanation should be discussed in the manuscript.

In Study 1, authors conclude that social norm about eating does not explain the behavioural shift (line 187 onward) because subjective apprehensiveness did not explain the decrease in the visible pair condition. Was subjective apprehensiveness compared also between conditions, i.e. did it differ between subjects in visible and invisible conditions? I think that it is important to clarify this because even if apprehensiveness did not explain the results within condition, it might still explain the observed differences between them.

Minor comments:

Statistical analyses are explained in detail in Supplementary material, but I think that it would be useful to include a short description of the analyses in the main manuscript. This would help readers to follow without needing to check Supplementary files.

Line 20: 'The foraging behavior' is probably a more appropriate wording

Line 26-27: I think that the use of 'taste test' (without further explanation) is confusing here because this was not the aim of the study. Perhaps it would be better to simply call it 'experiment' or something similar?

Line 47-48: Here authors discuss classical psychological studies with humans, but the cited literature includes experiments with fish, chicks and rats. Are these the right references?

Line 94: Please remove 'some' (all subjects in that condition were separated by the partition)

Line 101-103: The prediction here is that chicks in Visible pair condition would exhibit smaller number of grains per peck. As mentioned above, I believe that chicks never took more than one grain per peck? If so, could you please re-word this to avoid confusion.

Line 136-138: I understood that each subject had two trays (at the ends of the lane). Could you please clarify this here.

Line 165: Please remove 'other'

Line 166: Please remove 'also'

Line 202: I would suggest presenting all results from Study 1 before moving to Study 2, i.e. moving result (d) at the end of the results section.

Review form: Reviewer 2

Is the manuscript scientifically sound in its present form?

No

Are the interpretations and conclusions justified by the results?

No

Is the language acceptable?

Yes

Do you have any ethical concerns with this paper?

No

Have you any concerns about statistical analyses in this paper?

Yes

Recommendation?

Reject

Comments to the Author(s)

This study by Ogura et al examine how the presence of another forager influences an individual's foraging behaviour in humans and chicks. They tests this by conducting experimental foraging trials where subjects were either eating alone or in the presence of a second individual who was also eating their results show that in the presence of another individual forager select smaller food items more often. They argue this is a social tactic to reduce the potential for having food stolen by others since smaller food items less easy to steal.

This study addresses an interesting question, however I have a number of concerns about the conclusions it draws and the way it is designed. These are detailed below with references to specific line numbers but in particular:

- The authors draw conclusions from their study about social foraging behaviour across gregarious species. However, as their study only includes two species (humans and chicks) with a relatively small sample size for chicks (n = 9 and 8 in two experimental conditions) I do not think such general conclusions are justified. Indeed, the chick experiment seems far less developed than the human experiment (is has a smaller sample size and is missing one of the conditions included in the human experiment). I think the more scientifically sound conclusions from this study would be to focus on the human results (and even remove the chick data altogether) and focus on what they tell us about human social behaviour.

- Considerably more explanation is needed for the rationale behind the study design. For example, why are there three session and why are humans asked to rate the crisps' taste? Similarly, information is needed in the main text about the statistical methods used as well as explanation for the variables included in the statistical models.

Overall I think there is a scientifically sound study here with the potential to make an interesting contribution. However, a considerable amount of changes are needed to demonstrate this.

MAJOR COMMENTS

Line 45-47: “There have been few studies of how individuals in a group adjust their foraging tactics depending on the social context”. I don’t think this statement is accurate – there is wealth of literature examining this question. For example: Held et al (2010) *Animal Behaviour* 79, 857-862; Lee & Cowlshaw (2017) *Peer J* 5, e3462; Beauchamp (2013) *Biology Letters* 9, 2130528. These are just a few examples – the thorough search of the social foraging literature would identify many more studies.

Line 91-95: I think the rationale for the Invisible condition needs considerably more explanation. For example, why is this considered an intermediate level of competition? In addition, why was this included in the human trials but not in the chick trials (lines 96-103)?

Line 112-114: Why were there three sessions? The rationale for this is not explained and the difference between these sessions is not commented on or discussed later in the paper (though session number is controlled for in the analyses and the sessions are presented in in the figures, e.g. Figure 1A). I think this rationale needs to be explained more explicitly.

Line 118-120: Why was this rating phase included in the experiment? The data from these ratings are not presented in the results as far as I can see. I think perhaps it was to conceal the true motivation of the study from human subjects but this needs to be more clearly explained.

Line 125-133: The sample size of 2 to produce these baseline data is problematic. The experimental data are directly compared to these baseline data (figure 4C) yet the measurement of this baseline is likely to be inaccurate given this small sample size. I think the authors need to either (i) collect more baseline data, (ii) more explicitly acknowledge the potential unreliability of these analyses using baseline data, (iii) remove these analyses from the study.

Methods: No information is provided about the statistical methods used to analyse the data. This is crucial to allow the reader to understand what the study did, and to assess its scientific robustness. This is provided in the supplementary information (though this is not specified in the main text) but needs to be described in the main text of the manuscript.

Supplementary Information, lines 104-121: What is the rationale for including subject sex, and many interactions involving it, in the model? There are no predictions made about subject sex. I can see how including sex as a main effect would be sensible to control for sex-differences in foraging behaviour (though this rationale needs to be clearly stated) but the rationale for the many interactions is less clear to me.

Lines 274-304: Much of the discussion and argument presented by the authors is based on the premise that the foraging behaviour shift in the presence of a co-eater that they show in their results is potentially a general phenomenon across all gregarious species. However, I do not think the results support drawing such general conclusions. The results present data from two species (humans and chicks), with a relatively small sample size in one of these species (chicks, $n = 9$ and 8 in the two conditions, see line 135). I do not think generalising across all gregarious species from a sample size of two (with relatively little data on one species) is appropriate. At the very least the authors should acknowledge these limitations more explicitly. However, (as I highlight in my general comments above) I think the study might be more scientifically sound if the chick data were removed and conclusions were drawn about human social foraging behaviour rather than all gregarious species foraging behaviour.

MINOR COMMENTS

Line 26: I am not sure that describing humans as “intellectual” is appropriate or particularly informative. Humans have greater cognitive abilities than other animals in some areas, but other animals have greater cognitive skills in other aspects.

Line 73-103: A lot of the detail about the experimental procedure here is repeated in the Methods section. I understand the need to provide some detail here to help the reader understand the predictions, however I think some of the methodological detail presented here could be removed or summarised to avoid so avoid so much repetition.

Line 142: What does CCD stand for?

Decision letter (RSOS-191598.R0)

17-Dec-2019

Dear Dr Ogura:

Manuscript ID RSOS-191598 entitled "Mere Presence of Co-eater Automatically Shifts Foraging Tactics toward "Fast and Easy" Food in Humans and Domestic Chicks" which you submitted to Royal Society Open Science, has been reviewed. The comments from reviewers are included at the bottom of this letter.

In view of the criticisms of the reviewers, the manuscript has been rejected in its current form. However, a new manuscript may be submitted which takes into consideration these comments.

Please note that resubmitting your manuscript does not guarantee eventual acceptance, and that your resubmission will be subject to peer review before a decision is made.

Your resubmitted manuscript should be submitted by 15-Jun-2020. If you are unable to submit by this date please contact the Editorial Office.

on behalf of Dr Alecia Carter (Associate Editor) and Kevin Padian (Subject Editor)
openscience@royalsociety.org

Associate Editor Comments to Author (Dr Alecia Carter):

Dear authors,

I have now received two constructive and thoughtful reviews of your manuscript. As you will see, the reviewers found the study to be interesting, but both highlight some concerns with how

the data are interpreted and with the presentation of the information. I hope that you will find their comments will improve the manuscript.

Some very minor comments:

L68: resource competition is a universal problem requiring an adaptive solution, the problem itself is not adaptive.

L70: notion -> hypothesis

L75: see -> determine

L117: how was the scale invisible?

Reviewers' Comments to Author:

Reviewer: 1

Comments to the Author(s)

This study investigates how the presence of another individual influences foraging behaviour in humans and domestic chicks. The main finding is that both species reach smaller amounts of food more frequently when a conspecific is present, and this is suggested to result from the perceived risk of competition. The manuscript is well written and easy to read, and the statistical analyses are appropriate. After getting feedback from their previous submission to Proceedings of the Royal Society B, authors have conducted an additional test that shows that the choice pattern of humans in the presence of another person deviates from random sampling. This makes the conclusions about the shift towards smaller amounts of food stronger in Study 1. However, I think that there are still some issues (listed below) that should be addressed before publication.

One of the main conclusions of the paper is that humans and chicks show identical behavioural patterns, however, I'm not convinced that Study 1 and 2 are totally comparable. In contrast to human subjects that could take different amounts of chips per reach, I assume that chicks never took more than one grain per peck? Fig. 3c shows the probability of taking a grain per peck, which suggests that the presence of a conspecific mainly reduced chicks' likelihood to take a grain, rather than shifting their strategy to reaching smaller amounts of food. I therefore think that conclusions about parallel results in two studies (and setup) should be toned down. To be able to better compare the two species, future studies could present chicks with grains of different sizes to see if the presence of a conspecific shifts their preferences towards smaller food items.

The main result from Study 2 is that the peck frequency of chicks increased in the presence of a conspecific, and authors interpret this to be a result of competition. However, an alternative explanation is that as social species, chicks in solitary condition may have been more stressed and vigilant and therefore spent less time foraging. The presence of a companion chick might have reduced this stress and witnessing others foraging could have facilitated the same behaviour in observers, therefore increasing peck frequency. I think that this alternative explanation should be discussed in the manuscript.

In Study 1, authors conclude that social norm about eating does not explain the behavioural shift (line 187 onward) because subjective apprehensiveness did not explain the decrease in the visible pair condition. Was subjective apprehensiveness compared also between conditions, i.e. did it differ between subjects in visible and invisible conditions? I think that it is important to clarify this because even if apprehensiveness did not explain the results within condition, it might still explain the observed differences between them.

Minor comments:

Statistical analyses are explained in detail in Supplementary material, but I think that it would be useful to include a short description of the analyses in the main manuscript. This would help readers to follow without needing to check Supplementary files.

Line 20: 'The foraging behavior' is probably a more appropriate wording

Line 26-27: I think that the use of 'taste test' (without further explanation) is confusing here because this was not the aim of the study. Perhaps it would be better to simply call it 'experiment' or something similar?

Line 47-48: Here authors discuss classical psychological studies with humans, but the cited literature includes experiments with fish, chicks and rats. Are these the right references?

Line 94: Please remove 'some' (all subjects in that condition were separated by the partition)

Line 101-103: The prediction here is that chicks in Visible pair condition would exhibit smaller number of grains per peck. As mentioned above, I believe that chicks never took more than one grain per peck? If so, could you please re-word this to avoid confusion.

Line 136-138: I understood that each subject had two trays (at the ends of the lane). Could you please clarify this here.

Line 165: Please remove 'other'

Line 166: Please remove 'also'

Line 202: I would suggest presenting all results from Study 1 before moving to Study 2, i.e. moving result (d) at the end of the results section.

Reviewer: 2

Comments to the Author(s)

This study by Ogura et al examine how the presence of another forager influences an individual's foraging behaviour in humans and chicks. They tests this by conducting experimental foraging trials where subjects were either eating alone or in the presence of a second individual who was also eating their results show that in the presence of another individual forager select smaller food items more often. They argue this is a social tactic to reduce the potential for having food stolen by others since smaller food items less easy to steal.

This study addresses an interesting question, however I have a number of concerns about the conclusions it draws and the way it is designed. These are detailed below with references to specific line numbers but in particular:

- The authors draw conclusions from their study about social foraging behaviour across gregarious species. However, as their study only includes two species (humans and chicks) with a relatively small sample size for chicks ($n = 9$ and 8 in two experimental conditions) I do not think such general conclusions are justified. Indeed, the chick experiment seems far less developed than the human experiment (is has a smaller sample size and is missing one of the conditions included in the human experiment). I think the more scientifically sound conclusions from this study would be to focus on the human results (and even remove the chick data altogether) and focus on what they tell us about human social behaviour.

- Considerably more explanation is needed for the rationale behind the study design. For example, why are there three session and why are humans asked to rate the crisps' taste? Similarly, information is needed in the main text about the statistical methods used as well as explanation for the variables included in the statistical models.

Overall I think there is a scientifically sound study here with the potential to make an interesting contribution. However, a considerable amount of changes are needed to demonstrate this.

MAJOR COMMENTS

Line 45-47: "There have been few studies of how individuals in a group adjust their foraging tactics depending on the social context". I don't think this statement is accurate – there is wealth of literature examining this question. For example: Held et al (2010) *Animal Behaviour* 79, 857-862; Lee & Cowlshaw (2017) *Peer J* 5, e3462; Beauchamp (2013) *Biology Letters* 9, 2130528. These are just a few examples – the thorough search of the social foraging literature would identify many more studies.

Line 91-95: I think the rationale for the Invisible condition needs considerably more explanation. For example, why is this considered an intermediate level of competition? In addition, why was this included in the human trials but not in the chick trials (lines 96-103)?

Line 112-114: Why were there three sessions? The rationale for this is not explained and the difference between these sessions is not commented on or discussed later in the paper (though session number is controlled for in the analyses and the sessions are presented in the figures, e.g. Figure 1A). I think this rationale needs to be explained more explicitly.

Line 118-120: Why was this rating phase included in the experiment? The data from these ratings are not presented in the results as far as I can see. I think perhaps it was to conceal the true motivation of the study from human subjects but this needs to be more clearly explained.

Line 125-133: The sample size of 2 to produce these baseline data is problematic. The experimental data are directly compared to these baseline data (figure 4C) yet the measurement of this baseline is likely to be inaccurate given this small sample size. I think the authors need to either (i) collect more baseline data, (ii) more explicitly acknowledge the potential unreliability of these analyses using baseline data, (iii) remove these analyses from the study.

Methods: No information is provided about the statistical methods used to analyse the data. This is crucial to allow the reader to understand what the study did, and to assess its scientific robustness. This is provided in the supplementary information (though this is not specified in the main text) but needs to be described in the main text of the manuscript.

Supplementary Information, lines 104-121: What is the rationale for including subject sex, and many interactions involving it, in the model? There are no predictions made about subject sex. I can see how including sex as a main effect would be sensible to control for sex-differences in foraging behaviour (though this rationale needs to be clearly stated) but the rationale for the many interactions is less clear to me.

Lines 274-304: Much of the discussion and argument presented by the authors is based on the premise that the foraging behaviour shift in the presence of a co-eater that they show in their results is potentially a general phenomenon across all gregarious species. However, I do not think the results support drawing such general conclusions. The results present data from two species (humans and chicks), with a relatively small sample size in one of these species (chicks, $n = 9$ and 8 in the two conditions, see line 135). I do not think generalising across all gregarious species from a sample size of two (with relatively little data on one species) is appropriate. At the very least the authors should acknowledge these limitations more explicitly. However, (as I highlight in my general comments above) I think the study might be more scientifically sound if the chick data were removed and conclusions were drawn about human social foraging behaviour rather than all gregarious species foraging behaviour.

MINOR COMMENTS

Line 26: I am not sure that describing humans as “intellectual” is appropriate or particularly informative. Humans have greater cognitive abilities than other animals in some areas, but other animals have greater cognitive skills in other aspects.

Line 73-103: A lot of the detail about the experimental procedure here is repeated in the Methods section. I understand the need to provide some detail here to help the reader understand the predictions, however I think some of the methodological detail presented here could be removed or summarised to avoid so much repetition.

Line 142: What does CCD stand for?

Author's Response to Decision Letter for (RSOS-191598.R0)

See Appendix A.

RSOS-200044.R0

Review form: Reviewer 1

Is the manuscript scientifically sound in its present form?

Yes

Are the interpretations and conclusions justified by the results?

Yes

Is the language acceptable?

Yes

Do you have any ethical concerns with this paper?

No

Have you any concerns about statistical analyses in this paper?

No

Recommendation?

Accept with minor revision (please list in comments)

Comments to the Author(s)

This study investigates how the presence of a co-eater affects foraging strategies in humans. Compared to a previous submission, authors have now excluded a second study that was conducted with chicks, and this makes the manuscript more coherent. Authors have also adequately responded to other comments and revised the manuscript accordingly. The methods are now sufficiently described, results and their interpretation are clear, and the manuscript is well written. I have only few comments before publication.

Weighing experiment (line 243 onward):

I'm not sure how useful these results are because of the small sample size of only two subjects. I also think that the benefit of having a treatment that excludes eating would be to compare the effect of a conspecific's presence on reaching patterns when there is no competition for food. However, because the weighing experiment was conducted in a non-social situation, it does not allow this comparison and I would therefore suggest removing this result.

Other minor comments:

Line 45: Please change this to "such as on the effects"

Line 47: Please change this to "main scope has been on the effects"

Line 48: "focused on the effects"

Line 70: Please change this to "a previous study with domestic chicks"

Line 73: "a co-forager was present" might be a more appropriate wording

Line 111: Could you please include here the sample size of each sex.

Line 150: Please remove "the" (in all models)

Line 173: "were higher"

Line 181: "in the first session"

Line 182: "a saturation effect"

Line 184: What do you mean by other items? Other flavours or qualities?

Line 186: Please change "did not predict" to "predicted"

Dataset: Please remove chick data from the supporting data to avoid confusion.

Review form: Reviewer 2

Is the manuscript scientifically sound in its present form?

Yes

Are the interpretations and conclusions justified by the results?

Yes

Is the language acceptable?

Yes

Do you have any ethical concerns with this paper?

No

Have you any concerns about statistical analyses in this paper?

No

Recommendation?

Accept with minor revision (please list in comments)

Comments to the Author(s)

The authors have done a good job of addressing my concerns. In particular, I think the removal of the chick data make the study a more focused and robust piece of work that is likely to make a sound contribution. I have one concern that I feel has not been fully addressed.

Interactions included in the analyses: In the authors' response to my query about why so many interactions were included in their analyses they state that these were included as they were expecting the effect of their explanatory variables to differ between the sexes. They then highlight lines 176-178 and 181-182 as where they have detailed this rationale in the paper. These highlighted lines are both in the Results section and really just state that these interactions did not have an effect, e.g. line 176-178: "However, the interaction sex \times condition effect was not observed in either reach frequency or total food intake. That is, increase in reach frequency was commonly observed among men and women".

The extra statistical detail the authors have provided in the Methods is welcome but I feel this could be added to by fully stating what fixed effects, including interactions, were included in the analyses, and providing a clearer rationale for each of these. As it stands many tested effects are mentioned in the Results for the first time and this is a bit confusing. This needs to address all the interactions included. The authors mention two interactions in their response but there are more detailed in the supplementary material, e.g. four including a 3-way interaction in equation 2.

Decision letter (RSOS-200044.R0)

27-Feb-2020

Dear Dr Ogura

On behalf of the Editor, I am pleased to inform you that your Manuscript RSOS-200044 entitled "Mere Presence of Co-eater Automatically Shifts Foraging Tactics toward "Fast and Easy" Food in Humans" has been accepted for publication in Royal Society Open Science subject to minor revision in accordance with the referee suggestions. Please find the referees' comments at the end of this email.

The reviewers and Subject Editor have recommended publication, but also suggest some minor revisions to your manuscript. Therefore, I invite you to respond to the comments and revise your manuscript.

- Ethics statement

- Data accessibility

If you wish to submit your supporting data or code to Dryad (<http://datadryad.org/>), or modify your current submission to dryad, please use the following link:
<http://datadryad.org/submit?journalID=RSOS&manu=RSOS-200044>

- Competing interests

- Authors' contributions

AB carried out the molecular lab work, participated in data analysis, carried out sequence alignments, participated in the design of the study and drafted the manuscript; CD carried out the statistical analyses; EF collected field data; GH conceived of the study, designed the study,

coordinated the study and helped draft the manuscript. All authors gave final approval for publication.

- Acknowledgements

- Funding statement

Because the schedule for publication is very tight, it is a condition of publication that you submit the revised version of your manuscript before 07-Mar-2020. Please note that the revision deadline will expire at 00.00am on this date. If you do not think you will be able to meet this date please let me know immediately.

on behalf of Dr Alecia Carter (Associate Editor) and Kevin Padian (Subject Editor)
openscience@royalsociety.org

Associate Editor Comments to Author (Dr Alecia Carter):

Comments to the Author:

Dear authors,

Both of the original reviewers have now reviewed your revised manuscript and both are happy with the changes that were made. They now provide some very minor edits to improve clarity. However, although I agree with Reviewer 1's comment about the weighing experiment, the authors could consider presenting those data whilst acknowledging the limitations raised by reviewer 1 (and perhaps discussing what may have been a better design for this, as suggested by the reviewer).

At LL177-178, I wonder whether the interpretation of the non-significant sex \times condition interaction correct. Such an interaction isn't testing that the increase was observed in both men and women as suggested (that's the fixed effect of sex, with no interaction), but whether the increase was on the same magnitude for both men and women in the conditions. Men and women responded in the same way, but men were always higher than women in reach frequency. Or have I misunderstood (which is entirely possible)? In any case, this interpretation could be made clearer here with more careful wording.

I hope the authors find these minor corrections constructive.

Reviewer comments to Author:

Reviewer: 1

Comments to the Author(s)

This study investigates how the presence of a co-eater affects foraging strategies in humans. Compared to a previous submission, authors have now excluded a second study that was conducted with chicks, and this makes the manuscript more coherent. Authors have also adequately responded to other comments and revised the manuscript accordingly. The methods are now sufficiently described, results and their interpretation are clear, and the manuscript is well written. I have only few comments before publication.

Weighing experiment (line 243 onward):

I'm not sure how useful these results are because of the small sample size of only two subjects. I also think that the benefit of having a treatment that excludes eating would be to compare the effect of a conspecific's presence on reaching patterns when there is no competition for food. However, because the weighing experiment was conducted in a non-social situation, it does not allow this comparison and I would therefore suggest removing this result.

Other minor comments:

Line 45: Please change this to "such as on the effects"

Line 47: Please change this to "main scope has been on the effects"

Line 48: "focused on the effects"

Line 70: Please change this to "a previous study with domestic chicks"

Line 73: "a co-forager was present" might be a more appropriate wording

Line 111: Could you please include here the sample size of each sex.

Line 150: Please remove "the" (in all models)

Line 173: "were higher"

Line 181: "in the first session"

Line 182: "a saturation effect"

Line 184: What do you mean by other items? Other flavours or qualities?

Line 186: Please change "did not predict" to "predicted"

Dataset: Please remove chick data from the supporting data to avoid confusion.

Reviewer: 2

Comments to the Author(s)

The authors have done a good job of addressing my concerns. In particular, I think the removal of the chick data make the study a more focused and robust piece of work that is likely to make a sound contribution. I have one concern that I feel has not been fully addressed.

Interactions included in the analyses: In the authors' response to my query about why so many interactions were included in their analyses they state that these were included as they were expecting the effect of their explanatory variables to differ between the sexes. They then highlight lines 176-178 and 181-182 as where they have detailed this rationale in the paper. These highlighted lines are both in the Results section and really just state that these interactions did not have an effect, e.g. line 176-178: "However, the interaction sex \times condition effect was not observed in either reach frequency or total food intake. That is, increase in reach frequency was commonly observed among men and women".

The extra statistical detail the authors have provided in the Methods is welcome but I feel this could be added to by fully stating what fixed effects, including interactions, were included in the analyses, and providing a clearer rationale for each of these. As it stands many tested effects are mentioned in the Results for the first time and this is a bit confusing. This needs to address all the interactions included. The authors mention two interactions in their response but there are more detailed in the supplementary material, e.g. four including a 3-way interaction in equation 2.

Author's Response to Decision Letter for (RSOS-200044.R0)

See Appendix B.

Decision letter (RSOS-200044.R1)

10-Mar-2020

Dear Dr Ogura,

It is a pleasure to accept your manuscript entitled "Mere Presence of Co-eater Automatically Shifts Foraging Tactics toward "Fast and Easy" Food in Humans" in its current form for publication in Royal Society Open Science.

on behalf of Dr Alecia Carter (Associate Editor) and Kevin Padian (Subject Editor)
openscience@royalsociety.org

Appendix A

Associate Editor Comments to Author (Dr Alecia Carter):

Dear authors,

I have now received two constructive and thoughtful reviews of your manuscript. As you will see, the reviewers found the study to be interesting, but both highlight some concerns with how the data are interpreted and with the presentation of the information. I hope that you will find their comments will improve the manuscript.

Some very minor comments:

L68: resource competition is a universal problem requiring an adaptive solution, the problem itself is not adaptive.

We have changed the sentence as suggested (Line 76-77).

L70: notion -> hypothesis

We have deleted the description about chick study entirely in the revised manuscript.

L75: see -> determine

We have replaced the term with “examine” (Line 82).

L117: how was the scale invisible?

We are sorry about the unclarity. The scale was hidden by black cardboard (Line 123).

Reviewers' Comments to Author:

Reviewer: 1

Comments to the Author(s)

This study investigates how the presence of another individual influences foraging behaviour in humans and domestic chicks. The main finding is that both species reach smaller amounts of food more frequently when a conspecific is present, and this is suggested to result from the perceived risk of competition. The manuscript is well written and easy to read, and the statistical analyses are appropriate. After getting feedback from

their previous submission to Proceedings of the Royal Society B, authors have conducted an additional test that shows that the choice pattern of humans in the presence of another person deviates from random sampling. This makes the conclusions about the shift towards smaller amounts of food stronger in Study 1. However, I think that there are still some issues (listed below) that should be addressed before publication.

One of the main conclusions of the paper is that humans and chicks show identical behavioural patterns, however, I'm not convinced that Study 1 and 2 are totally comparable. In contrast to human subjects that could take different amounts of chips per reach, I assume that chicks never took more than one grain per peck? Fig. 3c shows the probability of taking a grain per peck, which suggests that the presence of a conspecific mainly reduced chicks' likelihood to take a grain, rather than shifting their strategy to reaching smaller amounts of food. I therefore think that conclusions about parallel results in two studies (and setup) should be toned down. To be able to better compare the two species, future studies could present chicks with grains of different sizes to see if the presence of a conspecific shifts their preferences towards smaller food items.

The comparability of the chick study with the human study was also questioned by Reviewer 2. As you pointed out, different from the human study where taking different amounts of chips per reach was possible, chicks never took more than one grain per peck. We agree with you that this aspect makes it difficult to compare the two studies directly. We have thus decided to remove Study 2 (chick study) entirely in the revised version, according to the suggestion of Reviewer 2.

The main result from Study 2 is that the peck frequency of chicks increased in the presence of a conspecific, and authors interpret this to be a result of competition. However, an alternative explanation is that as social species, chicks in solitary condition may have been more stressed and vigilant and therefore spent less time foraging. The presence of a companion chick might have reduced this stress and witnessing others foraging could have facilitated the same behaviour in observers, therefore increasing peck frequency. I think that this alternative explanation should be discussed in the manuscript.

As stated above, we have removed the chick study entirely from the manuscript.

However, the point you raised is very stimulating. Isolated chicks may have been vigilant and paid less attention to foraging per se. We would like to address this issue in a future study.

In Study 1, authors conclude that social norm about eating does not explain the behavioural shift (line 187 onward) because subjective apprehensiveness did not explain the decrease in the visible pair condition. Was subjective apprehensiveness compared also between conditions, i.e. did it differ between subjects in visible and invisible conditions? I think that it is important to clarify this because even if apprehensiveness did not explain the results within condition, it might still explain the observed differences between them.

In the *Visible Pair* condition, subjective apprehensiveness did *not* explain the decrease in the weight of potato chips per reach; the 95% credible interval of the coefficient of score for apprehensiveness included 0 at all timepoints (figure 2C; 1st to 23rd reach). In contrast, in the *Invisible Pair* condition, higher apprehensiveness was related with smaller weight of potato chips per reach in earlier reaches (figure 2C; 1st to 14th reach). Thus, the effect of “evaluation apprehension” was different between the *Visible Pair* condition and the *Invisible Pair* condition, i.e. subjective apprehensiveness partially explained the decrease in the weight of potato chips in the *Invisible Pair* condition, but not in the *Visible Pair* condition. Thus, we believe that social norm might have been at work partially, but could not explain the entire behavioral patterns (more frequent reach for smaller food) observed in this experiment. Please see Line 218-228.

Minor comments:

Statistical analyses are explained in detail in Supplementary material, but I think that it would be useful to include a short description of the analyses in the main manuscript. This would help readers to follow without needing to check Supplementary files.

As suggested, we have included the description of statistical analysis in the main text (Line 140-158).

Line 20: 'The foraging behavior' is probably a more appropriate wording

We have replaced "eating" with "foraging" (Line 20).

Line 26-27: I think that the use of 'taste test' (without further explanation) is confusing here because this was not the aim of the study. Perhaps it would be better to simply call it 'experiment' or something similar?

In the experiment, we used the word "taste test" to minimize the possibility that subjects explicitly felt a sense of competition or rivalry about eating. We have clarified this point in the revised manuscript (Line 27-28).

Line 47-48: Here authors discuss classical psychological studies with humans, but the cited literature includes experiments with fish, chicks and rats. Are these the right references?

We are sorry about this mistake. We have removed "with humans" (Line 49).

Line 94: Please remove 'some' (all subjects in that condition were separated by the partition)

We have removed "some" (Line 101).

Line 101-103: The prediction here is that chicks in Visible pair condition would exhibit smaller number of grains per peck. As mentioned above, I believe that chicks never took more than one grain per peck? If so, could you please re-word this to avoid confusion.

Line 136-138: I understood that each subject had two trays (at the ends of the lane). Could you please clarify this here.

As stated earlier, we have removed the chick study entirely from the manuscript.

Line 165: Please remove 'other'

Line 166: Please remove 'also'

We have removed the words as your recommendation (Line 191-192).

Line 202: I would suggest presenting all results from Study 1 before moving to Study 2,

i.e. moving result (d) at the end of the results section.

As stated earlier, we have removed the chick study entirely from the manuscript.

Reviewer: 2

Comments to the Author(s)

This study by Ogura et al examine how the presence of another forager influences an individual's foraging behaviour in humans and chicks. They test this by conducting experimental foraging trials where subjects were either eating alone or in the presence of a second individual who was also eating their results show that in the presence of another individual forager select smaller food items more often. They argue this is a social tactic to reduce the potential for having food stolen by others since smaller food items less easy to steal.

This study addresses an interesting question, however I have a number of concerns about the conclusions it draws and the way it is designed. These are detailed below with references to specific line numbers but in particular:

- The authors draw conclusions from their study about social foraging behaviour across gregarious species. However, as their study only includes two species (humans and chicks) with a relatively small sample size for chicks ($n = 9$ and 8 in two experimental conditions) I do not think such general conclusions are justified. Indeed, the chick experiment seems far less developed than the human experiment (is has a smaller sample size and is missing one of the conditions included in the human experiment). I think the more scientifically sound conclusions from this study would be to focus on the human results (and even remove the chick data altogether) and focus on what they tell us about human social behaviour.

Thank you for raising this important point. The incomparability between the chick study and the human study was also pointed out by Reviewer 1. We have thus decided to remove the chick study entirely and just focus on the human results.

- Considerably more explanation is needed for the rationale behind the study design. For example, why are there three sessions and why are humans asked to rate the crisps' taste? Similarly, information is needed in the main text about the statistical methods used as well as explanation for the variables included in the statistical models.

In the experiment, we used the word "taste test" to minimize the possibility that subjects explicitly felt a sense of competition or rivalry about eating (Line 27-28, Line 82-86). To examine whether saturation may affect eating in the presence of another co-eater, we repeated the test sessions three times in the experiment (Line 117-120).

We have also included the description of statistical analysis in the main text (Line 140-158), as suggested by Reviewer 1 as well.

Overall I think there is a scientifically sound study here with the potential to make an interesting contribution. However, a considerable amount of changes are needed to demonstrate this.

MAJOR COMMENTS

Line 45-47: "There have been few studies of how individuals in a group adjust their foraging tactics depending on the social context". I don't think this statement is accurate – there is wealth of literature examining this question. For example: Held et al (2010) *Animal Behaviour* 79, 857-862; Lee & Cowlishaw (2017) *Peer J* 5, e3462; Beauchamp (2013) *Biology Letters* 9, 2130528. These are just a few examples – the thorough search of the social foraging literature would identify many more studies.

Thank you for the reference. We have cited these papers. However, as we see them, the main scope of these studies was about effects of predation risk or social rank in group foraging, and is not the same as our focus in this paper (effects of resource competition per se). We have clarified this point in the revised manuscript. Please see Line 45-49.

Line 91-95: I think the rationale for the Invisible condition needs considerably more explanation. For example, why is this considered an intermediate level of competition? In addition, why was this included in the human trials but not in the chick trials (lines 96-103)?

In the *Invisible Pair* condition, pairs of subjects were separated by an opaque partition and were invisible (but audible) to each other during the test. We assumed that the partition, which separated the two subjects and the two food trays, should physically reduce the possibility of competition in the *Invisible Pair* condition, compared to the *Visible Pair* condition. We have also removed the chick study in the revised manuscript. Please see Line 98-105.

Line 112-114: Why were there three sessions? The rationale for this is not explained and the difference between these sessions is not commented on or discussed later in the paper (though session number is controlled for in the analyses and the sessions are presented in the figures, e.g. Figure 1A). I think this rationale needs to be explained more explicitly.

To examine whether saturation may affect eating in the presence of another co-eater, we repeated the test sessions three times. However, the model analysis revealed that participants reached for potato chips more frequently in the third session than in the first session, and no saturation effect was observed. Please see Line 117-120 and 179-182.

Line 118-120: Why was this rating phase included in the experiment? The data from these ratings are not presented in the results as far as I can see. I think perhaps it was to conceal the true motivation of the study from human subjects but this needs to be more clearly explained.

Thank you for raising this point. As you correctly guessed, we used the “taste test” setting to conceal the true motivation of the study and to prevent the subjects from having a sense of rivalry during the experiment (Line 27-28 and Line 117). We have also added the description of the rating results in the revised manuscript. Please see Line 183-188. Figure is shown in Supplementary figure 1.

Line 125-133: The sample size of 2 to produce these baseline data is problematic. The experimental data are directly compared to these baseline data (figure 4C) yet the measurement of this baseline is likely to be inaccurate given this small sample size. I think the authors need to either (i) collect more baseline data, (ii) more explicitly acknowledge the potential unreliability of these analyses using baseline data, (iii) remove

these analyses from the study.

According to your advice (ii), we have acknowledged the potential limitation of this analysis due to small sample size (Line 256-258). The unreliability of the baseline data is reflected in broader posterior predictive distribution shown in the far right panel of figure S3(B) (“*Weighing*”).

Methods: No information is provided about the statistical methods used to analyse the data. This is crucial to allow the reader to understand what the study did, and to assess its scientific robustness. This is provided in the supplementary information (though this is not specified in the main text) but needs to be described in the main text of the manuscript.

According to your and Reviewer 1’s suggestion, we have added the description of statistical methods (Line 140-158).

Supplementary Information, lines 104-121: What is the rationale for including subject sex, and many interactions involving it, in the model? There are no predictions made about subject sex. I can see how including sex as a main effect would be sensible to control for sex-differences in foraging behaviour (though this rationale needs to be clearly stated) but the rationale for the many interactions is less clear to me.

As you pointed out, previous studies have shown that social effect on eating behavior could be quite different by sex (Bellisle et al 1999; Vartanian et al 2007). Furthermore, we also had three sessions in the experiment to examine possible saturation effect (the effect of repeated sessions). We thus included these interaction effects specifically (sex × condition; session × condition) in the model. Please see Line 176-178, 181-182 of the main text.

Lines 274-304: Much of the discussion and argument presented by the authors is based on the premise that the foraging behaviour shift in the presence of a co-eater that they show in their results is potentially a general phenomenon across all gregarious species. However, I do not think the results support drawing such general conclusions. The results present data from two species (humans and chicks), with a relatively small sample size in one of these species (chicks, n = 9 and 8 in the two conditions, see line 135). I do not

think generalising across all gregarious species from a sample size of two (with relatively little data on one species) is appropriate. At the very least the authors should acknowledge these limitations more explicitly. However, (as I highlight in my general comments above) I think the study might be more scientifically sound if the chick data were removed and conclusions were drawn about human social foraging behaviour rather than all gregarious species foraging behaviour.

Thank you for raising this important point. We agree with you that our previous argument about generalizability across gregarious species was inappropriate. We have thus removed the chick study entirely and presented the human results only in the revised manuscript.

MINOR COMMENTS

Line 26: I am not sure that describing humans as “intellectual” is appropriate or particularly informative. Humans have greater cognitive abilities than other animals in some areas, but other animals have greater cognitive skills in other aspects.

We have removed this part in the revised manuscript.

Line 73-103: A lot of the detail about the experimental procedure here is repeated in the Methods section. I understand the need to provide some detail here to help the reader understand the predictions, however I think some of the methodological detail presented here could be removed or summarised to avoid so much repetition.

As suggested, we have reduced repeated descriptions of methodological details.

Line 142: What does CCD stand for?

As we have removed Study 2 (chick study), the term “CCD” has also been removed.

Appendix B

Comments to the Author:

Dear authors,

Both of the original reviewers have now reviewed your revised manuscript and both are happy with the changes that were made. They now provide some very minor edits to improve clarity. However, although I agree with Reviewer 1's comment about the weighing experiment, the authors could consider presenting those data whilst acknowledging the limitations raised by reviewer 1 (and perhaps discussing what may have been a better design for this, as suggested by the reviewer).

As suggested, we present the weighing data whilst acknowledging the limitation raised by reviewer 1 (Line 258-260).

At LL177-178, I wonder whether the interpretation of the non-significant sex \times condition interaction correct. Such an interaction isn't testing that the increase was observed in both men and women as suggested (that's the fixed effect of sex, with no interaction), but whether the increase was on the same magnitude for both men and women in the conditions. Men and women responded in the same way, but men were always higher than women in reach frequency. Or have I misunderstood (which is entirely possible)? In any case, this interpretation could be made clearer here with more careful wording.

I hope the authors find these minor corrections constructive.

As you say, *sex \times condition* interaction tested whether the increase was on the same magnitude for both men and women in the conditions. We have corrected the sentence (Line 178-179).

Reviewer comments to Author:

Reviewer: 1

Comments to the Author(s)

This study investigates how the presence of a co-eater affects foraging strategies in humans. Compared to a previous submission, authors have now excluded a second study that was conducted with chicks, and this makes the manuscript more coherent. Authors have also adequately responded to other comments and revised the manuscript accordingly. The methods are now sufficiently described, results and their interpretation are clear, and the manuscript is well written. I have only few comments before publication.

Weighing experiment (line 243 onward):

I'm not sure how useful these results are because of the small sample size of only two subjects. I also think that the benefit of having a treatment that excludes eating would be to compare the effect of a conspecific's presence on reaching patterns when there is no competition for food. However, because the weighing experiment was conducted in a non-social situation, it does not allow this comparison and I would therefore suggest removing this result.

According to the editor's suggestion, we present the weighing data whilst acknowledging the limitation (Line 258-260).

Other minor comments:

Line 45: Please change this to "such as on the effects"

Line 47: Please change this to "main scope has been on the effects"

Line 48: "focused on the effects"

Line 70: Please change this to "a previous study with domestic chicks"

Line 73: "a co-forager was present" might be a more appropriate wording

We have changed the sentences as suggested (Lines 45, 47, 48, 70, and 73, respectively).

Line 111: Could you please include here the sample size of each sex.

We have included the sample size of each sex (Line 108-110).

Line 150: Please remove "the" (in all models)

Line 173: "were higher"

Line 181: "in the first session"

Line 182: "a saturation effect"

We have changed the sentences as suggested (Lines 151, 174, 182 and 188, respectively).

Line 184: What do you mean by other items? Other flavours or qualities?

We mean other flavors and textures by "other items". Please see Line 186.

Line 186: Please change "did not predict" to "predicted"

We have changed the sentence as suggested (Line 188).

Dataset: Please remove chick data from the supporting data to avoid confusion.

We removed chick data from the depository.

Reviewer: 2

Comments to the Author(s)

The authors have done a good job of addressing my concerns. In particular, I think the removal of the chick data make the study a more focused and robust piece of work that is likely to make a sound contribution. I have one concern that I feel has not been fully addressed.

Interactions included in the analyses: In the authors' response to my query about why so many interactions were included in their analyses they state that these were included as they were expecting the effect of their explanatory variables to differ between the sexes. They then highlight lines 176-178 and 181-182 as where they have detailed this rationale in the paper. These highlighted lines are both in the Results section and really just state that these interactions did not have an effect, e.g. line 176-178: "However, the interaction $sex \times condition$ effect was not observed in either reach frequency or total food intake. That is, increase in reach frequency was commonly observed among men and women". The extra statistical detail the authors have provided in the Methods is welcome but I feel this could be added to by fully stating what fixed effects, including interactions, were included in the analyses, and providing a clearer rationale for each of these. As it stands many tested effects are mentioned in the Results for the first time and this is a bit confusing. This needs to address all the interactions included. The authors mention two interactions in their response but there are more detailed in the supplementary material, e.g. four including a 3-way interaction in equation 2.

We have addressed the other two interactions ($sex \times session$ and $sex \times condition \times session$). The result indicates that a saturation effect was not observed. Please see Line 182-183.